# OVERCOMING BIAS TOWARDS BASE SESSIONS IN FEW-SHOT CLASS-INCREMENTAL LEARNING (FSCIL)

## ABSTRACT

Few-shot class-incremental learning (FSCIL) with a more realistic and challenging problem setting aims to learn a set of novel object classes with a restricted number of training examples in sequence. In the process, striking a balance between not forgetting previously-learned object classes and overfitting to novel ones plays a crucial role. Meanwhile, conventional methods exhibit a significant performance bias towards a base session: excessively low incremental performance compared to base performance. To tackle this, we propose a simple-but-effective pipeline that achieves a substantial performance margin for incremental sessions. Further, we devise and perform comprehensive experiments under diverse conditions—leveraging pretrained representations, various classification modules, and aggregation of the predictions within our pipeline; our findings reveal essential insights towards model design and future research directions. Additionally, we introduce a set of new evaluation metrics and benchmark datasets to address the limitations of the conventional metrics and benchmark datasets which disguise the bias towards a base session. These newly introduced metrics and datasets allow to estimate the generalization of FSCIL models. Furthermore, we achieve new state-of-the-art performance with significant margins as a result of our study. The codes of our study are available at GITHUB.

## 1 INTRODUCTION

Deep learning has revolutionized various computer vision tasks such as image classification (Deng et al., 2009; Tang et al., 2015), object detection (Girshick, 2015; Amit et al., 2020), instance segmentation (He et al., 2017; Bolya et al., 2019) and visual relation detection (Xu et al., 2017; Kim et al., 2020); they have become a standard approach in multiple fundamental areas. However, conventional learning methodologies assume the availability of a fixed set of target classes in advance and a large-scale dataset for effective training. Therefore, these methodologies hardly deal with novel classes for intelligent agents deployed in real-world environments. In real-world settings, learning methodologies need to handle novel classes with a restricted number of training samples.

The above-mentioned issue naturally leads to the task of class incremental learning (CIL) (Mittal et al., 2021). CIL aims to update a classifier encountering novel classes in an incremental manner. One of the challenges of CIL is striking a balance between catastrophic forgetting of previously-learned classes and overfitting to novel classes. Moreover, the few-shot class-incremental learning (FSCIL) task (Tao et al., 2020) has recently emerged to formulate CIL in a more practical and challenging setting—the focus of our work. FSCIL attempts to incrementally learn novel classes by utilizing very few training samples without forgetting formerly learned classes over multiple incremental sessions.

In this study, we uncover a critical issue in current FSCIL methods. They tend to achieve significantly higher base accuracy compared to incremental accuracy—leading to a notable performance imbalance. As Fig. 1 shows, current state-of-the-art (SoTA) FSCIL methods yield poor performance when evaluated solely on each incremental session. Thus, current FSCIL methods exhibit challenges in adapting to novel classes—not aligning with the fundamental objective of FSCIL. Especially, the commonly-used metric in FSCIL, the average accuracy on all the encountered classes, has a bias towards a base session having the largest number of test samples (Peng et al., 2022); high performance on base session classes can easily inflate the metric value and overshadow the underlying

problems of poor adaptation to incremental sessions and the presence of performance imbalance. Furthermore, conventional datasets bear analogous characteristics to ImageNet and the inductive bias of convolution would lead to unexpected negative consequences; potentially biasing the model design process. Therefore, we claim the standard evaluation benchmark is not thorough enough to reveal such consequences and results in poor adaptation to incremental sessions.

To address these issues, we propose a novel pipeline, a set of new metrics, and new benchmark datasets for FSCIL. First, we develop a simple-yet-effective pipeline for FSCIL that can effectively learn incremental classes—mitigating the performance bias towards a base session. The pipeline consists of three key components: 1) feature extraction, 2) classification module, and 3) prediction aggregation. Each component leverages pretrained representations, various classification modules, and appropriate aggregation of the predictions, respectively. In order to disclose essential insights for the construction of each component, we conduct comprehensive experiments; our meticulously-designed experiments provide invaluable guidance for designing a strong FSCIL model. Moreover, the proposed pipeline demonstrates superior session-level generalization compared to current SoTA methods.

Next, we introduce a set of novel performance metrics that allow granular session-level evaluations. Specifically, our metrics assess the accuracy of each session individually and quantify the extent of performance bias towards a base session. Unlike the conventional metric, our metrics represent how effectively an FSCIL method generalizes over novel classes and attains balanced performance between base and incremental sessions. Furthermore, we collect two new benchmark datasets for evaluation in the process of designing our study. These datasets facilitate the evaluation of model robustness across various target distributions and the propensity of CNN architectures to adapt to different target distributions, thereby aiding in the development of unbiased model design guidelines.

In summary, our main contributions are as follows:

- **Mitigation of bias**: We mitigate the prevalent performance bias issue in FSCIL with the proposed simple pipeline comprised of three essential components: 1) feature extraction 2) classification module and 3) prediction aggregation.
- **Empirical study design**: We devise a set of extensive empirical studies to reveal important insights regarding the three components of our pipeline.
- **Evaluation metrics**: We design evaluation metrics that focus on session-level adaptation and the degree of performance bias.
- **Benchmark datasets**: We construct two additional benchmark datasets for in-depth investigation of the effectiveness of FSCIL methods.
- **SoTA performance**: Our pipeline not only mitigates the bias but also achieves the SoTA performance in terms of both conventional and proposed session-level metrics.

## 2 RELATED WORK

### 2.1 CLASS INCREMENTAL LEARNING (CIL)

CIL pursues to learn the knowledge of new object classes from a sequence of datasets without forgetting the knowledge of previously-learned object classes (Zhu et al., 2021). CIL methods attempt to address the issue of *catastrophic forgetting* of previous knowledge and *overfitting* to new knowledge. In the context of computer vision, three main categories constitute CIL approaches. The first category makes use of a knowledge distillation loss to address catastrophic forgetting (Rebuffi et al., 2017; Li & Hoiem, 2018), and the second category utilizes a replay buffer for storing representative samples of previous classes (Welling, 2009; Belouadah & Popescu, 2019) and the third category designs a model parameter selection scheme (Zenke et al., 2017; Kirkpatrick et al., 2017).

### 2.2 FEW-SHOT CLASS INCREMENTAL LEARNING (FSCIL)

The FSCIL task (Tao et al., 2020) investigates the CIL task in a more practical and challenging setting; an FSCIL method needs to handle novel classes with an extremely small number of training samples (less than 10 samples per class) although the method can utilize abundant training data for

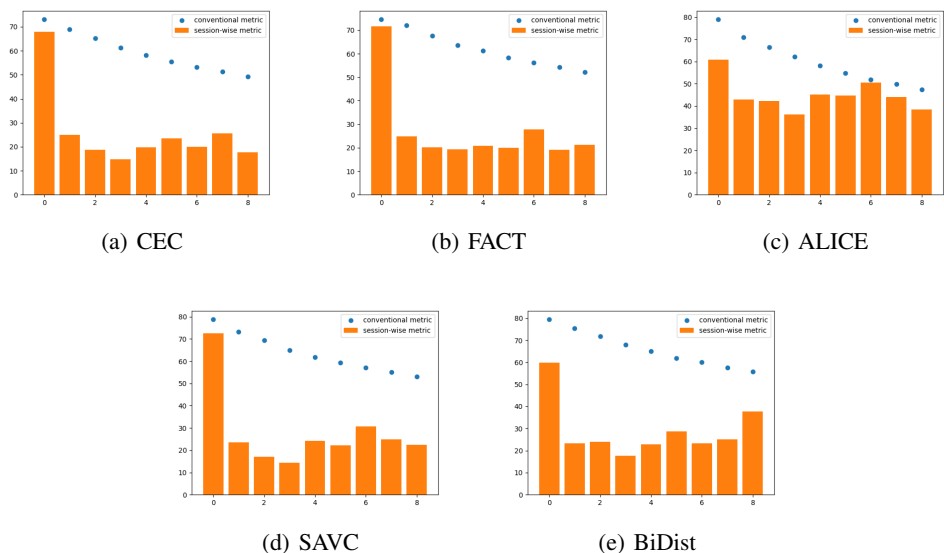

<table>
<tr><td>(a) CEC</td><td>(b) FACT</td><td>(c) ALICE</td></tr>
<tr><td>(d) SAVC</td><td>(e) BiDist</td><td></td></tr>
</table>

Figure 1: The comparison between the proposed session-wise accuracy and conventional metric for each previous method on CIFAR100 (the horizontal axis represents session numbers; 0 for the base session). The incremental performance is far below satisfactory in contrast to the high base session accuracy, which is obscured by the conventional metric.

each object class in the base session. Various research efforts have led to the advancement of FS-CIL. The beginning work on FSCIL has proposed the TOPIC framework which employs a neural gas network to preserve the topology of features across previous and new object classes (Tao et al., 2020). CEC has employed a pseudo incremental learning paradigm and a graph neural network to learn context information for classifiers (Zhang et al., 2021). C-FSCIL has exploited a meta-learning mechanism for learning quasi-orthogonal feature representations and additional memory for prototypes of each class (Hersche et al., 2022). MetaFSCIL has devised a learning method based on meta-learning for fast adaptation and prevention of forgetting in incremental sessions (Chi et al., 2022). FACT has concentrated on forward compatibility and utilized virtual prototypes and virtual instances for securing room for incremental classes in the feature embedding space (Zhou et al., 2022). AL-ICE has made use of the angular penalty loss (CosFace) (Wang et al., 2018), data augmentation, and class augmentation techniques for learning both discriminative and transferable features (Peng et al., 2022). FeSSSS (Ahmad et al., 2022) has leveraged self-supervised feature representations and a Gaussian generator to deal with overfitting and catastrophic forgetting (Ahmad et al., 2022). SAVC (Song et al., 2023) has aimed to achieve better separation between base classes and virtual classes generated by predefined transformations to capture diverse information. BiDist (Zhao et al., 2023) has presented a distillation structure with two teachers,each alleviating overfitting and forgetting, respectively.

In spite of the effectiveness of conventional FSCIL methods, they happen to focus on improving base session performance and display unsatisfactory performance for each incremental session, which gets hidden by the conventional performance metrics.

## 2.3  PRETRAINED REPRESENTATIONS

**Self-Supervised Learning (SSL).** Self-supervised representation learning obtains labels from the input data using semi-automatic processes and predicts part of the data from other parts (Liu et al., 2021a)—removing the need for manual annotation. Thus, SSL could leverage the massive amount of unlabeled data for feature representation learning. SSL methods fall into either generative or discriminative approaches although there exist other categorizations. Generative approaches require substantial computations and a number of discriminative approaches, especially contrastive approaches, are currently displaying state-of-the-art performance (Grill et al., 2020). Among recent

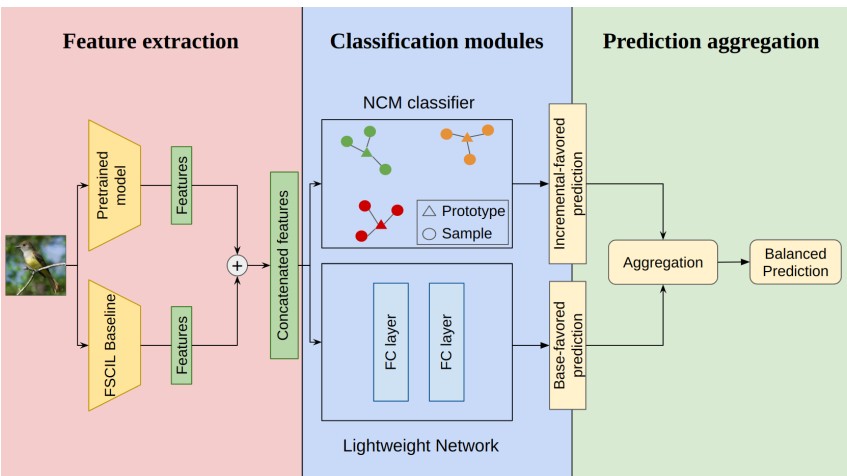

Figure 2: Overall workflow of our proposed pipeline. It consists of three key components: 1) feature extraction 2) classification modules, and 3) prediction aggregation. In the Feature extraction, we concatenate the features extracted from the baseline and pretrained backbone. Subsequently, the concatenated features go into the classification modules, and their predictions are aggregated.

advanced SSL methods, DeepCluster-v2 (Caron et al., 2018; 2020) clusters previously-learned representations for updating new representations, SeLa-v2 (Asano et al., 2019; Caron et al., 2020) combines clustering and representation learning through principled learning formulation that averts degeneracy, SwAV (Caron et al., 2020) exploits contrastive methods without requiring to compute pairwise comparisons, Moco-v2 (Chen et al., 2020b) implements the design improvements of SimCLR (Chen et al., 2020a) in the MoCo framework (He et al., 2020), MoBY (Xie et al., 2021) tunes advances in SSL towards ViTs (Han et al., 2022), and DINO (Caron et al., 2021) reveals the properties of ViTs learned by self-distillation with no labels.

**Transfer Learning.** Transfer learning transfers the knowledge learned from a source task to a target task as the name suggests (Pan & Yang, 2010). Transfer learning helps tackle the data deficiency problem of the target task by exploiting rich feature representations from the source task where large-scale training datasets are generally available. Various transfer learning techniques have emerged such as fine-tuning (Wang et al., 2017b; Kolesnikov et al., 2020; Kumar et al., 2022), knowledge distillation (Park et al., 2021; Zhao et al., 2022; Beyer et al., 2022) and domain adaptation (Gopalan et al., 2011; Tommasi & Caputo, 2013; Jin et al., 2020). Furthermore, studies on the transferability of pretrained representations are recently drawing increasing attention (Ericsson et al., 2021; Islam et al., 2021; Pándy et al., 2022; Yang et al., 2023). Our work is in line with these studies in that we investigate the transferability of pretrained representations for FSCIL.

## 3 METHOD

In this section, we described the proposed pipeline for FSCIL that alleviates the bias toward a base session and achieves high performance in both base and incremental sessions. Fig. 2 depicts the overall pipeline.

### 3.1 PROBLEM FORMULATION

The FSCIL task includes two types of learning sessions: 1) a single base session and 2) a set of multiple incremental sessions.

**Base Session**: In the base session, an FSCIL method takes in the training dataset $\mathcal{D}_{\text{train}}^0 = \{(\mathbf{x}_i^0, y_i^0)\}_{i=1}^{|D_{\text{train}}^0|}$, where $\mathbf{x}_i^0$ and $y_i^0$ represent the input, *i.e.*, an image, and the ground-truth object class label, respectively. The training dataset of the base session delivers a *sufficient* num-

ber of training samples and the FSCIL method measures its performance on the test dataset $\mathcal{D}_{\text{test}}^0 = \{(\mathbf{x}_i^0, y_i^0)\}_{i=1}^{|D_{\text{test}}^0|}$.

**Incremental Sessions**: After the base session, the FSCIL method undergoes a set of incremental sessions in sequence; the FSCIL method accepts a series of datasets $\{\mathcal{D}^1, ..., \mathcal{D}^s, ..., \mathcal{D}^N\}$, where $\mathcal{D}^s = (\mathcal{D}_{\text{train}}^s, \mathcal{D}_{\text{test}}^s)$ representing training and test datasets and $N$ denotes the total number of incremental sessions. There is no overlapping labels between the label sets of object classes for incremental sessions, *i.e.*, $\mathcal{C}^i \cap \mathcal{C}^j = \varnothing$ for $\forall i, j$ and $i \neq j$, where $\mathcal{C}^s$ stands for the set of class labels of $s$-th session. Moreover, the training datasets of consecutive incremental sessions encompass *insufficient* numbers of training data samples. The $N$-way $K$-shot setting illuminates that each incremental session deals with $N$ object classes with $K$ training samples per class. The performance evaluation protocol after the last incremental session handles entire object classes that have been considered over all sessions $\mathcal{C}^0 \cup \mathcal{C}^1 \cup ... \cup \mathcal{C}^N$.

## 3.2 Feature extraction: leveraging pretrained representations

### 3.2.1 Motivation and process

In the context of FSCIL, training the feature extractor only in the base session is typically preferred since incremental sessions have small amount of data. However, exclusive training of the feature extractor in the base session exhibits a critical limitation: poor novel class adaptation, which results in a significant performance drop for incremental sessions. To address this, we propose to leverage recent advancements in pretrained representations. We note that instead of utilizing excessively large datasets for pretraining, which can lead to overly simplistic performance improvement, we employ ImageNet, akin to previous methods when training on CUB200 [1].

One simple idea for utilizing pretrained representations is fine-tuning. However, updating model parameters in incremental sessions would lead to a detrimental impact on base performance with fine-tuning. On the other hand, freezing representations for the whole sessions would restrict plasticity. To cope with this dilemma, our architecture leverages two types of features from a frozen pretrained backbone and an FSCIL baseline through concatenation (Ahmad et al., 2022).

### 3.2.2 Design space of feature extraction

**Backbone architectures.**

- CNN: We utilize the extensively-studied ResNet50 backbone architecture (He et al., 2016).
- ViT: We employ the DeiT-small (DeiT-s) (Touvron et al., 2021) and Swin Transformer-tiny (Swin-t) (Liu et al., 2021b) backbones which have similar numbers of parameters as ResNet50. Further, ViT could have different patch sizes and we denote it as the number at the end such as DeiT-s8 and DeiT-s16.

**Learning methodologies for pretraining.**

- *Supervised learning (SL)*: Supervised representations have extracted features from ImageNet (Russakovsky et al., 2015) using the CrossEntropy loss.
- *Self-supervised learning (SSL)*: Self-supervised representations learn from semi-automatic processes and various methods exist. For CNN, we examine DeepCluster-v2 (Caron et al., 2018), Moco-v2 (Chen et al., 2020b), SeLa-v2 (Asano et al., 2019; Caron et al., 2020), SwAV (Caron et al., 2020) and DINO (Caron et al., 2021); for ViT, DINO (Caron et al., 2021) and MoBY (Xie et al., 2021).

## 3.3 Classification module: How can we make the most of the extracted representations?

To maximize the utility of extracted representations, we employ a classification module after feature extraction. Moreover, three widely-used categories of classification modules are available for struc-

---

[1]Current methods initiate training from ImageNet-pretrained representations due to relatively small amount of base session data in CUB200

turing our pipeline. Previous studies, to the best of our knowledge, have predominantly focused on the utilization of a single classifier type from the three categories, with limited exploration of their potential synergies. In contrast, our pipeline exploits these categories of classification modules both individually and in combination. The three categories are as follows:

**Lightweight network (LN)** (Ahmad et al., 2022): LN is a 2-layer multi-layer perceptron (MLP) and gets updated by the usual back-propagation. To overcome the catastrophic forgetting, old class instances are sampled from the corresponding gaussian distribution estimated on each session.

**Meta-learning-based classifier (GAT)** (Zhang et al., 2021): Among various types of meta-learning-based classifiers for FSCIL, we involve a meta-learned Graph Attention Network (Veličković et al., 2017) to refine the classifier trained for each session.

**Nearest class mean (NCM)** (Hou et al., 2019): Metric-based classifier with cosine similarity is another commonly-used classification module in FSCIL. Since it is non-parametric, it is less susceptible to overfitting, especially in scenarios with limited data such as incremental sessions. We employ the nearest class mean (NCM) classifier with balanced testing (Peng et al., 2022).

### 3.4 PREDICTION AGGREGATION: ENHANCING BOTH BASE AND INCREMENTAL PERFORMANCE

The majority of previous methods have endeavored to attain high performance using a single classifier—leading to complicated yet biased algorithms. However, achieving favorable performance in both base and incremental sessions solely with a single module poses significant challenges due to the plasticity-stability dilemma (Abraham & Robins, 2005).

As a solution, we propose to decouple the classifier into two distinct entities and use the aggregation of predictions from each entity as the overall prediction. The two entities are as follows: one designed for base-favored prediction and the other for incremental (inc)-favored prediction. The base-favored module shows relatively high performance in base sessions compared to the inc-favored module which outperforms the base-favored counterpart in predicting incremental classes. As these modules can complement each other, the aggregation of their predictions can yield a synergistic effect. We perform the aggregation by simply adding the softmax output of each module. Since LN and GAT tend to yield significantly sharper distribution than NCM, we use a temperature (Wu et al., 2018) to control the peakiness of the output.

## 4 EVALUATION BENCHMARK

### 4.1 DATASETS

In our experiments, we consider three datasets[2]: one widely-used FSCIL dataset (CUB200 (Wah et al., 2011)) and two proposed datasets (Flowers102 and DCC). We do not include the *mini*ImageNet (Russakovsky et al., 2015) dataset for fair comparison since we utilize ImgaeNet-pretrained representations in our experiments and the dataset is a subset of ImageNet—resulting in meaninglessly-superb performance. Moreover, we propose two additional benchmark datasets since conventional datasets display analogous distributions as ImageNet (Russakovsky et al., 2015) and would result in biased results (Ericsson et al., 2021). Thus, we construct two FSCIL evaluation benchmarks using datasets that exhibit distinctive distributions from ImageNet (see Fig. 3).

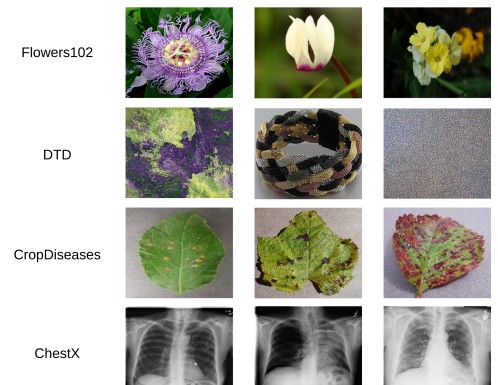

Figure 3: Examples of the new benchmark datasets. Each row displays images corresponding to the dataset indicated on the leftmost labels.

---

[2]For experiments on CIFAR100 (Krizhevsky & Hinton, 2009), refer to the supplementary material

Table 1: Experimental results for individual classification modules. Best results are marked in bold and second best results are underlined.

| Method | CUB200 | | | | Flowers102 | | | | DCC | | | |
|---|---|---|---|---|---|---|---|---|---|---|---|---|
| | $a_0$ | Inc$_{avg}$ | IMB | A$_s$ | $a_0$ | Inc$_{avg}$ | IMB | A$_s$ | $a_0$ | Inc$_{avg}$ | IMB | A$_s$ |
| LN | **81.91** | 57.74 | 24.17 | 59.94 | **97.84** | 90.84 | 7.00 | 91.62 | 82.31 | 44.70 | 37.61 | 50.97 |
| GAT | 79.85 | 60.76 | 19.09 | 62.50 | 97.42 | 90.46 | 6.96 | 91.23 | **82.69** | 44.80 | 37.89 | 51.11 |
| NCM | 74.62 | **62.22** | **12.39** | **63.35** | 93.67 | **92.85** | **0.82** | **92.94** | 77.99 | **54.50** | 23.49 | **58.41** |

**CUB200.** CUB200 initially designed for fine-grained image classification provides $224 \times 224$ sized 11,788 images of 200 object classes. For incremental learning, we partition the 200 object classes into 100 base session classes and 100 incremental session classes to configure a 10-way 5-shot setting following the standard evaluation protocol (Tao et al., 2020).

**Flowers102.** Based on the Oxford 102 Folowers dataset (Nilsback & Zisserman, 2008), we build the Flowers102 dataset. We use the first 100 classes of the dataset excluding the last two classes. Instead of using the original training-test split, we transfer some data from the test set to the training set due to the limited number of data available for each class. Specifically, we set the number of base classes to 60 and transfer 20 images per base class from the test set to the training set, resulting in approximately 40 training images per class. Next, we construct 8 incremental sessions in a 5-way 5-shot manner, and secure approximately 20 test images per class. Finally, we resize the images to $224 \times 224$ following the standard.

**DTD-CropDiseases-ChestX (DCC).** We construct the DCC dataset combining DTD (Cimpoi et al., 2014), CropDiseases (Mohanty et al., 2016) and ChestX (Wang et al., 2017a). One thing to note is CropDiseases and ChestX are two of the benchmark datasets of the cross-domain few-shot learning (CD-FSL) (Guo et al., 2020)—they make DCC significantly different from ImageNet. We organize the DCC dataset in the way that each session includes classes from all three datasets (DTD, CropDiseases, and ChestX). Particularly, the base session consists of 67 classes (37 from DTD, 28 from CropDiseases, and 2 from ChestX) with 80 images per class. DCC contains 5 incremental sessions (5-way 5-shot). Moreover, we combine 2, 2, and 1 classes from each of the three datasets to gather 5 classes in each incremental session. For the test dataset, we collect 40 images per class and we resize the images to $224 \times 224$.

## 4.2 PERFORMANCE METRICS

**Limitations of Conventional Metric.** The most commonly used metric in FSCIL calculates top-1 accuracy for all sessions encountered up to the current session. However, the metric is biased towards the base session having the largest test samples, thereby hindering an accurate evaluation of adaptability to incremental sessions (Peng et al., 2022). Due to this biased metric, the low incremental performance of most current methods is often overshadowed.

**Proposed session-level evaluation metrics.** In order to conduct a more detailed evaluation, we propose to use the session-wise accuracy in FSCIL, *i.e.*, $i$-th session accuracy ($a_i$). $a_i$ independently assesses performance for each session at the end of the training and considers classes only in $\mathcal{C}^i$. In addition to the session-wise accuracy, we introduce additional metrics, *Incremental average* (Inc$_{avg}$), *Imbalance* (IMB), and *Session Average* (A$_s$) as follows:

$$\text{Inc}_{\text{avg}} = \frac{\sum_{i=1}^{N} a_i}{N}, \quad (1) \qquad \text{IMB} = a_0 - \text{Inc}_{\text{avg}}, \quad (2) \qquad \text{A}_s = \frac{\sum_{i=0}^{N} a_i}{N+1}, \quad (3)$$

where $a_i$ is $i$-th session accuracy and $N$ is the number of incremental sessions. Inc$_{avg}$ and IMB evaluate incremental class adaptation and the degree of the bias towards a base session, respectively. A$_s$ indicates the average of session-wise accuracy which ensures equal weighting for all sessions in our evaluation.

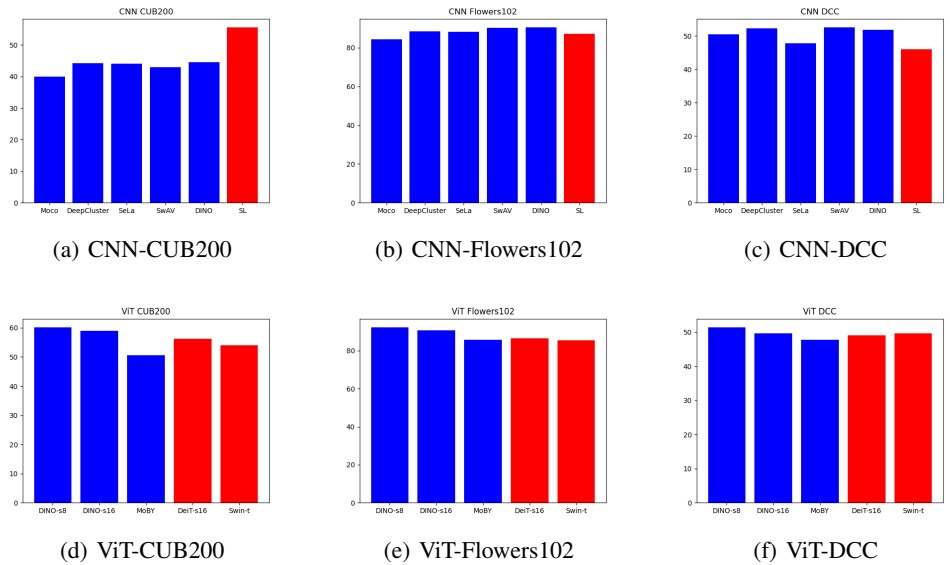

Figure 4: The average of session-wise accuracy, i.e. $A_s$, for various types of pretrained representations; blue indicating SSL representations while red SL representations. For the baseline, we employed ALICE (Peng et al., 2022).

Table 2: Experimental results for the combination of the classification modules. Best results are marked in bold and second best results are underlined.

| Method | CUB200 | | | | Flowers102 | | | | DCC | | | |
|---|---|---|---|---|---|---|---|---|---|---|---|---|
| | $a_0$ | $Inc_{avg}$ | IMB | $A_s$ | $a_0$ | $Inc_{avg}$ | IMB | $A_s$ | $a_0$ | $Inc_{avg}$ | IMB | $A_s$ |
| LN + GAT | **82.37** | 62.80 | 19.57 | 64.58 | **98.50** | 91.71 | 6.79 | 92.47 | **83.58** | 47.10 | 36.48 | 53.18 |
| LN + NCM | 79.54 | **63.77** | 15.77 | **65.20** | 96.42 | **94.11** | 2.31 | **94.37** | 80.97 | **55.00** | **25.97** | **59.33** |
| GAT + NCM | 75.94 | 62.50 | **13.44** | 63.72 | 93.75 | 92.85 | **0.90** | 92.95 | 80.00 | 53.20 | 26.80 | 57.67 |
| LN + NCM + GAT | 79.71 | 63.63 | 16.08 | 65.09 | 96.08 | 93.47 | 2.61 | 93.76 | 80.86 | 54.60 | 26.26 | 58.98 |

Table 3: Comparative study result. The proposed framework consistently outperforms baselines under various conditions. Best results are marked in bold and second best results are underlined.

| Method | CUB200 | | | | Flowers102 | | | | DCC | | | |
|---|---|---|---|---|---|---|---|---|---|---|---|---|
| | $a_0$ | $Inc_{avg}$ | IMB | $A_s$ | $a_0$ | $Inc_{avg}$ | IMB | $A_s$ | $a_0$ | $Inc_{avg}$ | IMB | $A_s$ |
| CEC (Zhang et al., 2021) | 71.30 | 27.47 | 43.84 | 31.45 | 96.00 | 62.46 | 33.54 | 66.19 | 76.67 | 29.85 | 46.82 | 37.65 |
| FACT (Zhou et al., 2022) | 72.84 | 39.78 | 33.06 | 42.78 | 96.71 | 77.45 | 19.26 | 79.59 | 79.85 | 26.64 | 53.21 | 35.51 |
| ALICE (Peng et al., 2022) | 68.44 | 51.65 | 16.79 | 53.17 | 93.25 | 84.71 | 8.54 | 85.66 | 76.53 | 44.90 | 31.63 | 50.17 |
| SAVC (Song et al., 2023) | 77.44 | 47.53 | 29.91 | 50.25 | **97.75** | 83.45 | 14.30 | 85.04 | **81.53** | 42.60 | 38.93 | 49.09 |
| BiDist (Zhao et al., 2023) | 71.44 | 45.66 | 25.78 | 48.00 | 95.67 | 82.70 | 12.97 | 84.14 | 75.78 | 48.60 | 27.18 | 53.13 |
| Ours | **79.54** | **63.77** | **15.77** | **65.2** | 96.42 | **94.11** | **2.31** | **94.37** | 80.97 | **55.00** | **25.97** | **59.33** |

## 5 EXPERIMENTS

### 5.1 ABLATIONS AND ANALYSIS

#### 5.1.1 ANALYSIS ON UTILIZATION OF PRETRAINED REPRESENTATIONS IN FSCIL

Fig. 4 shows $A_s$ for various pretrained representations on the proposed datasets. CNN-based representations display sensitivity to the target dataset's characteristics. For CUB200 which has analogous characteristics as ImageNet, SL ResNet50 attains the leading performance among other pretrained representations. On the other hand, SSL CNNs display superior performance than SL CNN for Flowers102 and DCC which entail distinctive characteristics compared to ImageNet. In this case, certain

SSL CNNs such as DeepCluster-v2, SwAV and DINO even surpass supervised ViTs. Notably, the performance gap between SSL and SL widens as the target distribution diverges further from that of ImageNet—the largest gap for DCC. In the case of ViT, they do not exhibit dependency on the distribution of the target dataset. Especially, DINO-s8 consistently outperforms other categories of pretrained representations.

### 5.1.2 ANALYSIS ON THE CLASSIFICATION MODULES

Table 1 contains the results of the session-level metrics for the LN, GAT, and NCM. When using LN and GAT, it achieves high base accuracy but low $Inc_{avg}$, whereas NCM demonstrates relatively lower base accuracy and higher $Inc_{avg}$. In other words, the prediction of LN, GAT can be considered as base-favored, while that of NCM can be seen as inc-favored for the prediction aggregation process.

### 5.1.3 ANALYSIS ON THE PREDICTION AGGREGATION

Given that LN, GAT can serve as a base-favored module and NCM as an inc-favored module, two aggregation scenarios are available: 1) LN+NCM and 2) GAT+NCM. Table 2 presents the possible combinations and shows that LN+NCM outperforms GAT+NCM. Specifically, LN+NCM enhances $Inc_{avg}$ compared to NCM, with only a slight decline in base accuracy compared to LN. NCM entailing a lower IMB is mainly due to its low base accuracy, which is also a crucial metric for FSCIL. Hence, in this context, we could prioritize achieving higher $A_s$ for an effective FSCIL module.

Conversely, GAT does not consistently display advantages in aggregation. For LN+GAT, the aggregation leads to improvements in all the session-level metrics compared to LN. Specifically, GAT increases $Inc_{avg}$, indicating that GAT predictions contain useful information for predicting incremental classes that complements LN. However, when comparing NCM and NCM+GAT in terms of $A_s$, there is a minimal difference or NCM even outperforms NCM+GAT on DCC. Moreover, LN+NCM consistently outperforms LN+NCM+GAT across all the datasets. Notably, GAT fails to effectively increase base accuracy and $Inc_{avg}$.

### 5.1.4 PIPELINE CONFIGURATION

Based on our analyses, we identified that DINO-s8 is powerful and robust representation for our feature extraction. In terms of the classification modules and prediction aggregation, our findings indicate that the aggregation of LN and NCM possess strong prediction power for base and incremental sessions—rendering the addition of GAT redundant. Consequently, employing DINO-s8 with LN+NCM is the optimal configuration for our pipeline which can effectively address the bias towards base session.

## 5.2 COMPARATIVE STUDY

Table 3 presents the comparative results between the previous methods and our pipeline using the proposed metrics. Our pipeline significantly outperforms the previous methods, achieving a new SoTA. Notably, our newly introduced session-level metrics uncover that baselines display poor adaptability to the incremental sessions (low $Inc_{avg}$) and $A_s$, which was obscured by the conventional metric. On the other hand, our pipeline effectively resolves these limitations of the previous methods and demonstrates substantial performance superiority.

## 6 CONCLUSION

In this study, we unveil the critical bias towards a base session in FSCIL. To address this, we propose a simple pipeline and introduce a set of novel session-level metrics as well as new benchmark datasets for meticulous analysis and evaluation of robustness to the target distribution shift. We conducted a comprehensive range of experiments and showed our pipeline's effectiveness in both mitigating the bias and facilitating novel class adaptation. It successfully achieves a superior balance between base and incremental performance while enhancing the overall performance on the new benchmark datasets as evidenced by the introduced metrics. We believe our proposed pipeline holds the potential to make a substantial contribution to the advancement of FSCIL algorithms.

## REPRODUCIBILITY STATEMENT

We have uploaded the code we used in our experiments to our GitHub repository. Detailed experimental settings are provided in the supplementary material. Comprehensive descriptions of the three components of our pipeline can be found in Section 3. With this information, reproduction of our study should be easily achievable.

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
