Table 4: Session-wise accuracy and $Inc_{avg}$ for the previous methods on CIFAR100. They exhibit poor incremental performance, which shows a significant bias towards the base sessions.

| Method | Sessions | | | | | | | | | $Inc_{avg}$ |
|---|---|---|---|---|---|---|---|---|---|---|
| | 0 | 1 | 2 | 3 | 4 | 5 | 6 | 7 | 8 | |
| CEC (Zhang et al., 2021) | 67.98 | 25.00 | 18.80 | 14.80 | 19.80 | 23.60 | 20.00 | 25.60 | 17.60 | 20.65 |
| FACT (Zhou et al., 2022) | 71.78 | 24.80 | 20.20 | 19.40 | 20.80 | 20.00 | 27.80 | 19.20 | 21.20 | 21.68 |
| ALICE (Peng et al., 2022) | 60.90 | 42.80 | 42.20 | 36.20 | 45.20 | 44.80 | 50.60 | 44.00 | **38.40** | 43.69 |
| SAVC (Song et al., 2023) | **72.58** | 23.60 | 17.00 | 14.40 | 24.20 | 22.20 | 30.60 | 24.80 | 22.40 | 22.40 |
| BiDist (Zhao et al., 2023) | 59.83 | 23.40 | 24.00 | 17.60 | 22.80 | 28.60 | 23.40 | 25.20 | 37.80 | 25.35 |
| Ours | 62.78 | **46.00** | **42.20** | **37.40** | **54.40** | **50.80** | **65.20** | **49.80** | 34.40 | **47.53** |

Table 5: Session-wise accuracy and $Inc_{avg}$ for the previous methods on miniImageNet. They exhibit poor incremental performance, which shows a significant bias towards the base sessions.

| Method | Sessions | | | | | | | | | $Inc_{avg}$ |
|---|---|---|---|---|---|---|---|---|---|---|
| | 0 | 1 | 2 | 3 | 4 | 5 | 6 | 7 | 8 | |
| CEC (Zhang et al., 2021) | 67.87 | 11.63 | 16.81 | 14.50 | 9.90 | 11.24 | 11.12 | 20.67 | 14.11 | **13.75** |
| FACT (Zhou et al., 2022) | 73.65 | 12.80 | 12.20 | 15.40 | 11.40 | 5.40 | 5.40 | 17.00 | 14.20 | **11.73** |
| ALICE (Peng et al., 2022) | 67.92 | 34.00 | 28.80 | 34.00 | 32.60 | 40.20 | 30.60 | 50.80 | 53.20 | **35.86** |
| SAVC (Song et al., 2023) | 77.52 | 23.80 | 19.40 | 24.00 | 23.80 | 10.40 | 13.80 | 26.40 | 29.80 | **21.43** |
| BiDist (Zhao et al., 2023) | 69.32 | 15.60 | 16.20 | 250 | 16.40 | 23.60 | 18.40 | 39.80 | 46.40 | **30.08** |

Table 6: Session-wise accuracy and $Inc_{avg}$ for the previous methods on CUB200. They exhibit poor incremental performance, which shows a significant bias towards the base sessions.

| Method | Sessions | | | | | | | | | | | $Inc_{avg}$ |
|---|---|---|---|---|---|---|---|---|---|---|---|---|
| | 0 | 1 | 2 | 3 | 4 | 5 | 6 | 7 | 8 | 9 | 10 | |
| CEC (Zhang et al., 2021) | 71.30 | 29.14 | 18.62 | 16.64 | 44.09 | 23.81 | 30.63 | 24.53 | 22.56 | 35.94 | 28.69 | 27.47 |
| FACT (Zhou et al., 2022) | 72.84 | 45.16 | 38.33 | 23.83 | 46.00 | 29.87 | 44.44 | 36.03 | 34.24 | 55.56 | 44.33 | 39.78 |
| ALICE (Peng et al., 2022) | 68.44 | **59.86** | 43.90 | 35.57 | 55.33 | 35.57 | 55.90 | 54.55 | 48.47 | 66.32 | 61.00 | 51.65 |
| SAVC (Song et al., 2023) | 77.44 | 46.24 | 42.51 | 35.91 | 56.00 | 34.23 | 50.35 | 46.13 | 43.73 | 62.85 | 57.33 | 47.53 |
| BiDist (Zhao et al., 2023) | 71.44 | 45.52 | 33.10 | 25.84 | 56.00 | 35.91 | 44.44 | 44.11 | 43.05 | 65.63 | 63.00 | 45.66 |
| Ours | **79.54** | 57.71 | **54.01** | **56.04** | **72.00** | **41.61** | **67.36** | **64.65** | **60.34** | **83.33** | **80.67** | **63.77** |

# A IMPLEMENTATION DETAIL

For training LN, we followed the training procedure of FeSSSS Ahmad et al. (2022). The training procedure involves a Gaussian generator and we used the vector model for the variance estimation. We empirically set the epoch for training in the base session to 300 through our experiments. We fixed the epoch for incremental sessions to 500. Hyperparameters such as batch size, learning rate, and optimizer were all set to the same as those of FeSSSS Ahmad et al. (2022). For NCM and GAT, we followed ALICE Peng et al. (2022) and CEC (Zhang et al., 2021), respectively. We employed ALICE (Peng et al., 2022) as the baseline and DINO-s8 as the pretrained representations for our pipeline. Experiments for previous methods on Flowers102 and DCC were conducted under the same conditions as those for CUB200. The CIFAR100 dataset Krizhevsky & Hinton (2009) provides images with the size of $32 \times 32$—though the pretrained models haved learned representations from $224 \times 224$ images. Thus, we employed ResNet20 pretrained with images with the size of $32 \times 32$ in a supervised manner for our feature extraction process on CIFAR100.

**Selection of the temperature for the prediction aggregation.** We set the temperature to maximize $A_s$. However, for GAT+NCM on DCC, increasing the temperature resulted in higher $A_s$, ultimately matching the performance of using NCM alone. Therefore, as an exception to show the detrimental effect of NCM in aggregation, we chose to use a temperature value similar to that for other datasets rather than maximizing $A_s$.

Table 7: Experiment results on CIFAR100 with the proposed session-level metrics for the previous methods. They show a significant bias to the base sessions.

| Method | CIFAR100 | | | |
|---|---|---|---|---|
| | $a_0$ | $Inc_{avg}$ | IMB | $A_s$ |
| CEC (Zhang et al., 2021) | 67.98 | 20.65 | 47.33 | 25.91 |
| FACT (Zhou et al., 2022) | 71.78 | 21.68 | 50.11 | 27.24 |
| ALICE (Peng et al., 2022) | 60.90 | 43.69 | 17.21 | 45.01 |
| SAVC (Song et al., 2023) | **72.58** | 22.40 | 50.18 | 27.98 |
| BiDist (Zhao et al., 2023) | 59.83 | 25.35 | 34.48 | 29.18 |
| Ours | 62.78 | **47.53** | **15.26** | **49.22** |

Table 8: Session-level analysis on various pretrained representations. We followed the same experimental conditions of FeSSSS (Ahmad et al., 2022).

| | Method | CUB200 | | | | Flowers102 | | | | DCC | | | |
|---|---|---|---|---|---|---|---|---|---|---|---|---|---|
| | | $a_0$ | $Inc_{avg}$ | IMB | $A_s$ | $a_0$ | $Inc_{avg}$ | IMB | $A_s$ | $a_0$ | $Inc_{avg}$ | IMB | $A_s$ |
| CNN | Moco-v2 | 74.55 | 36.56 | 37.99 | 40.01 | 98.17 | 82.68 | 15.48 | 84.40 | 82.28 | 44.20 | 38.08 | 50.55 |
| | DeepCluster-v2 | 77.03 | 40.89 | 36.13 | 44.18 | 97.58 | 87.34 | 10.25 | 88.47 | 82.65 | 46.20 | **36.45** | 52.27 |
| | SeLa-v2 | 75.56 | 40.93 | 34.63 | 44.08 | 97.08 | 87.09 | 10.00 | 88.20 | 81.46 | 41.00 | 40.46 | 47.74 |
| | SwAV | 76.78 | 39.58 | 37.09 | 42.96 | **98.50** | 89.21 | 9.29 | 90.24 | **83.32** | **46.40** | 36.92 | **52.55** |
| | DINO-ResNet50 | 77.65 | 41.20 | 36.46 | 44.51 | 98.33 | 89.46 | 8.87 | 90.45 | 82.84 | 45.60 | 37.24 | 51.81 |
| | Supervised | 79.82 | 53.11 | 26.71 | 55.54 | 98.42 | 85.69 | 12.73 | 87.10 | 82.28 | 38.70 | 43.58 | 45.96 |
| ViT | DeiT-s16 | 81.28 | 53.65 | 27.64 | 56.16 | 97.67 | 85.21 | 12.46 | 86.59 | 81.72 | 42.50 | 39.22 | 49.04 |
| | Swin-t | 81.46 | 51.33 | 30.13 | 54.07 | 97.83 | 83.81 | 14.02 | 85.37 | 82.54 | 43.10 | 39.44 | 49.67 |
| | DINO-s8 | **82.19** | **58.08** | **24.11** | **60.27** | 97.84 | **90.84** | **7.00** | **91.62** | 82.31 | 44.70 | 37.61 | 50.97 |
| | DINO-s16 | 82.16 | 56.55 | 25.60 | 58.88 | 98.50 | 89.84 | 8.66 | 90.80 | 82.54 | 43.10 | 39.44 | 49.67 |
| | MoBy-Swin-t | 76.78 | 48.01 | 28.77 | 50.62 | 97.67 | 84.08 | 13.59 | 85.59 | 80.86 | 41.20 | 39.66 | 47.81 |

## B    OTHER EXPERIMENT RESULTS

Tables 4, 5, 6 show the session-wise accuracy and $Inc_{avg}$ for CIFAR100, miniImageNet and CUB200. We note that for miniImageNet, we do not compare our pipeline against baselines since our pipeline leverages representations pretrained from ImageNet—resulting in meaninglessly-superb performance. The session-wise results indicate that the proposed pipeline displays superior session-wise accuracy across nearly every incremental session—corroborating that our pipeline achieves remarkable novel class adaptation.

Table 7 shows the experiment results on the CIFAR100 dataset. As in (5.2), our pipeline significantly outperforms the previous methods, achieving a new SoTA. Furthermore, our newly introduced session-level metrics reveal that baselines possess poor adaptability to the incremental sessions (low $Inc_{avg}$) and $A_s$, which was obscured by the conventional metric. On the other hand, our pipeline effectively surmounts these limitations of the previous methods and demonstrates substantial performance superiority.

Table 8 compares different types of pretrained representations and their effectiveness. CNN architectures exhibit vulnerability toward target distribution shifts as analyzed in (5.1.1). In addition, ViT-based representations bring about superior performance. Overall, the DINO-s8 representation brings about a good performance across various target distributions. Therefore, we opt for DINO-s8 as pretrained representation in our pipeline.