# OpenReview forum: "Overcoming bias towards base sessions in few-shot class-incremental learning (FSCIL)"
_ICLR.cc/2024/Conference — Submitted to ICLR 2024_

### Official Review · Reviewer_qGui · 2023-10-29

**Soundness:** 3 good
**Presentation:** 2 fair
**Contribution:** 2 fair
**Rating:** 5
**Confidence:** 5

**Summary:**

This paper addresses the problem of Few-Shot Class-Incremental Learning (FSCIL). FSCIL consists of two stages: a base session involving training on a large-scale base dataset and incremental sessions with a few-shot setting. The evaluation metric used in previous methods is the mean accuracy of all test samples, where performance is dominated by the base classes due to their larger number of test samples. The performance in the incremental sessions alone is much worse than in the base session. To mitigate this bias towards the base classes and balance the two learning stages, this paper proposes an investigation into the knowledge of pre-trained models and classifier types. A well-pre-trained model is utilized and kept frozen, along with an updating base model. Three types of classifiers are explored, and those demonstrating superior performance in both the base and incremental sessions are combined for further improvement. Additionally, new evaluation metrics are introduced to separately assess issues like forgetting and incremental performance. Two new benchmarks are also proposed.

**Strengths:**

- The dominance of base class accuracy as an evaluation metric for previous FSCIL methods is indeed unfair. The proposed evaluation metrics provide a more intuitive benchmark for evaluation.
- A baseline method can be significantly improved by simply utilizing and aggregating a frozen pretrained model.
- Combining various types of classifier heads has also been shown to be beneficial.

**Weaknesses:**

- The overall method seems less novel: utilizing the frozen pre-trained backbone to improve the forgetting issue is not new (e.g., [1]). The runtime increases for inference, while [1] does not as the knowledge merging happens in the parameter space; aggregating the prediction from multiple classifiers is an ensemble operation.
- Regarding lightweight network (LN), catastrophic forgetting is overcoming by replay the samples from previous classes? Such process violates the setting of FSCIL, where accessing the data from multiple incremental sessions is prohibited.
- What is the rationale behind choosing ALICE as the baseline model over other options? Is this choice based on heuristics or specific reasons?
- In the CUB200 benchmark, if the settings remain consistent with those in other papers, why do the previous methods listed in Table 3 exhibit significantly lower performance, even on the base classes (a_0), compared to their reported results in their original papers?
- The comparison with prior methods may not be entirely fair, as all of them exclusively use ResNet-18 or ResNet-20, which have significantly fewer parameters compared to ResNet-50 or ViT. Is the observed performance boost primarily attributed to the larger number of parameters?
- An essential baseline is missing: Would fine-tuning the pre-trained model on the base classes and using it as a new "frozen pre-trained model" lead to an overall performance improvement? In the current method pipeline, it assumes that the pre-trained model possesses knowledge highly correlated with each dataset. However, this assumption may be vulnerable.

Dataset:
- The two proposed benchmarks, Flower102 and DCC, may not be entirely suitable for FSCIL. The fundamental concept of FSCIL involves a large-scale dataset in the base session, which is reasonable for offline data collection and training. However, Flower102 and DCC have relatively small scales, with only 20 and 80 images per class for the base sessions, totaling 1200 and 5360 images, respectively. This can pose challenges for algorithm development, particularly as models grow larger. Additionally, the limited number of test images can result in significant evaluation variation


A missing prior work: Liu et al. "Few-Shot Class-Incremental Learning via Entropy-Regularized Data-Free Replay." ECCV 2022.

[1] Zhang et al. “Grow and Merge: A Unified Framework for Continuous Categories Discovery.” NeurIPS 2022.

**Questions:**

- In Table 8, are the experiments conducted excluding the baseline model as in Fig. 2?
- Should “Flowers102” be replaced by “Flowers100” as only the first 100 classes are used?

---

> ### Author Response · Authors · 2023-11-14
>
> We would like to express our sincere gratitude for the time and effort you dedicated to reviewing our paper. Our response to the weakness and questions is present below.
>
> Weakness_1.
>
> A: Although aggregating predictions falls under the category of ensemble operations, we claim that our method differs significantly from conventional ones. Unlike typical ensemble methods that do not specify particular types of classification modules, our pipeline employs two distinct types: the base-favored and incremental-favored modules (Sec 3.4). It is crucial to note that simply combining a large number of classification modules does not necessarily lead to improvement. This is because many current methods excel only in the base, and combining such methods does not address the persisting issue of performance bias. As we have proposed in Sec 5.1, incorporating a classification module favoring the incremental session is essential for achieving overall balanced performance.
>
> Weakness_2.
>
> A: When training LN, we do not replay samples from the previous sessions. As we have described in the supplementary material (A), we trained a Gaussian generator for each class in each session (which is inspired by the setting of FeSSSS) and subsequently sampled data from the generator for the respective class.
>
> Weakness_3.
>
> A: While SAVC and BiDistFSCIL are the latest approaches, the high IMB values in Table 3 indicate that their performance exhibits a pronounced bias toward the base session. This results in an inflated value of the biased conventional metric, misleadingly suggesting proficiency as FSCIL methods. Conversely, ALICE achieved the most balanced performance according to our proposed metrics. Therefore, we decided to use ALICE as our baseline.
>
> Weakness_4.
>
> A: The discrepancy arises from our evaluation using the newly proposed metrics. While previous approaches assess the performance for all classes encountered up to the current session after training at each session, we assess session-level adaptability by exclusively evaluating the classes relevant to a specific session only after training on the last session (Sec 4.2). Hence, our reported values differ from those provided in the original papers.
>
> Weakness_5.
>
> A: We acknowledge that the improvement in performance compared to existing methods can be attributed to the larger backbone. We would appreciate it if the focus could shift from such a perspective to ‘exploring how to effectively utilize backbones with a larger number of parameters (especially in the size constraint of ResNet50) in FSCIL.’ Tables 1 and 2 illustrate that when using a larger backbone, simply employing LN, GAT, or NCM individually results in a biased performance—either exhibiting high accuracy only in the base session (LN and GAT) or relatively higher incremental performance with a notable decline in base performance (NCM). We emphasize that integrating the two specific types of classification modules we proposed can effectively address this issue by achieving an optimal balance in performance compared to the naive use of a large backbone.
>
> Weakness_6.
>
> A: We assumed that fine-tuning the backbone with base classes induces an inevitable bias towards the base classes, a notion supported by the biased performance of current methods observed in our experiments. Such biased features may not only be unhelpful but could potentially be detrimental in adapting to the incremental sessions. Consequently, we posited the frozen pretrained backbone would be better rather than fine-tuned ones even if the pretrained knowledge is not highly correlated with the incremental sessions. While there may be instances where fine-tuning with base classes can be highly beneficial for adaptation to incremental sessions if there is a significant similarity between the distributions of base and incremental sessions, we believe such instances are generally rare. Therefore, we did not fine-tune the pretrained backbone with the base classes. However, as you suggested, using a fine-tuned backbone as a baseline could provide a more in-depth understanding of the impact of fine-tuning with base classes. We will carefully consider this advice.
>
> Weakness_7.
>
> A: From the perspective of the number of images per base class, (as we have described in sec 4.1,) the proposed datasets have more images—Flowers102 has 40, DCC has 80 images—than CUB200 where there are around 30 training images per base class. Furthermore, they can serve as valuable tools for assessing the robustness against the target distribution shifts. However, we agree that more test images would have been beneficial.
>
> Q1.
>
> A: Table 8 indicates the results evaluated with our proposed metrics where both baseline and pretrained backbones are used. The setup is the same as that of Figure 4.
>
> Q2.
>
> A: As you mentioned, since only the first 100 classes of Flowers102 are utilized, renaming it to Flowers100 seems appropriate to distinguish it from the original Flowers102.

---

> > ### Comment · Reviewer_qGui · 2023-11-20
> > **Response to authors**
> >
> > I would like to thank the authors for addressing my concerns. Some of them are well-addressed, therefore, I increase my rating from 3 to 5. But I still lean to reject the paper, mainly due to the concerns on novelty:
> >
> > The overall pipeline is to utilize an imagenet pre-trained model due to its better representation. Several classification methods on top of representation are examined per the proposed evaluation metric. The ones that perform better on each metric are then aggregated by ensemble the final logits. Although a better benchmark performance is achieved, the whole method is too heuristic and simple, which weighs more for making my review decision.

---

### Official Review · Reviewer_SuEP · 2023-11-01

**Soundness:** 2 fair
**Presentation:** 2 fair
**Contribution:** 2 fair
**Rating:** 5
**Confidence:** 2

**Summary:**

In this paper, authors find that conventional methods exhibit excessively low incremental  session performance compared to base session performance. To tackle this issue, authors propose a new pipeline and conduct comprehensive experiments explore various conditions, including pre-trained representations, classification modules, and prediction aggregation. This paper further introduces new evaluation metrics and benchmark datasets to mitigate bias towards the base session, allowing for a more accurate assessment of FSCIL model generalization.

**Strengths:**

1. Authors reveal the fact that current approaches cannot generalize on novel classes well.
2. Authors propose a novel pipeline, evaluation metrics and datasets for FSCIL.

**Weaknesses:**

Overall, though the paper might be useful, the paper lacks a clear motivation of the proposed methods and details of important parts of the model. This paper lacks the consistency between the efforts in the paper and conclusion. Meanwhile, the empirical results are not compelling enough to validate the effectiveness of proposed pipeline and evaluation metrics.

My questions and concerns are as follows:

1.	One important contribution of the paper is proposing the pipeline to mitigate the performance issue, however, there is no detailed analysis of the reason behind the phenomenon. Why the proposed pipeline is able to tackle this issue?

2.	The propose pipeline utilizes a pre-trained model to provide representations to aid FSCIL baseline. However, if the pre-trained model has already learned novel classes or similar classes, does it violate the incremental learning setting of learning novel classes?

3.	In section 3.1, the description of incremental sessions causes misunderstanding. N is used to represent both the number of incremental sessions and classes in each session.

4.	In section 3.4, details are missing on how to aggregate the predictions. If the model is able to obtain the classification results of base and novel classes, why further aggregates the two prediction results to get the final predicition?

5.	In table 3, the performance of a_0 of compared methods are listed wrong.  SAVC achieves 81.85 at a_0; FACT reports 75.90 at a_0; ALICE reports 77.40, etc. Also, the comparison might not be fair since majority of current methods adopt backbones like ResNet-12, 18, 20.

6.	Though author constructs two additional benchmark datasets for investigation of the effectiveness of FSCIL methods, as one important contribution, there is no explanation on the choice of two dataset, or comparisons between the proposed datasets and  previous benchmarks.

7.	In section 4.2, what does IMB mean? Why the performance margin between the base session and each incremental session can measure the bias towards the base session? Also, session average is commonly used metric in previous works, this cannot be included as an contribution.

8.	 In conclusion, “we propose a simple pipeline and introduce a set of novel session-level    metrics as well as new benchmark datasets for meticulous analysis and evaluation of robustness to the target distribution shift ”. I fail to see any related efforts on the evaluation of robustness to the target distribution shift, it seems irrelevant with the work of this paper.

**Questions:**

see the weakness box

---

> ### Author Response · Authors · 2023-11-14
>
> We would like to express our sincere gratitude for the time and effort you dedicated to reviewing our paper. Our response to the weakness and questions is present below.
>
> Weakness_1,4.
>
> A: Using only LN or NCM individually does not yield optimal outcomes. LN excels in the base session but falters in the incremental sessions, while NCM performs well in the incremental sessions but poorly in the base. Given their contrasting tendencies, we hypothesized that the two classification modules can potentially complement each other. Indeed, we observed that aggregating their output distributions significantly improved overall performance (Section 5.1).
>
> - The detailed process of the aggregation
> The aggregation process entails summing the output distributions from LN and NCM. However, we noticed LN's output distribution exhibits excessive sharpness compared to NCM, leading to LN’s dominance when simply summed.
> To address this, we smooth LN’s distribution with an optimal temperature [ref.1] before adding it to NCM's. Through multiple experiments, we find the optimal temperature, observing that LN's smoothened distribution slightly dominates when evaluating test data for the base session while NCM's has a slightly stronger influence when evaluation for the incremental sessions. Consequently, this aggregation enhances overall performance balance.
>
> Weakness_2,6.
>
> A: As we have described in Sec 4.1, we addressed the concern of the violation by setting the target distribution that diverges significantly from the pretraining dataset (ImageNet). Our criterion is as follows: if a self-supervised pretrained backbone yields better transferability to a target dataset than the supervised pretrained one, the target dataset has a considerable distribution shift from the pretraining dataset. This is primarily grounded in the fact that the supervised training aligns more closely with the distribution it trained with, due to the availability of label information.
>
> According to [ref.2], when the pretraining dataset is ImageNet, and the target dataset is Flowers102, DTD, or CD-FSL benchmarks (including CropDiseases and ChestX), self-supervised pretrained backbones outperform. Based on this, we chose Flowers102, DTD, and the CD-FSL benchmarks for the new datasets. However, due to limited data in DTD and CD-FSL benchmarks, we decided to combine them. Among the four datasets in CD-FSL, since EuroSAT has a much smaller resolution than the others, and including ISIC makes training challenging, we propose the new dataset DCC, which consists of DTD, CropDiseases, and ChestX. Unlike previous benchmarks similar to ImageNet, these datasets can introduce greater diversity to the benchmarks.
>
> Weakness_3.
>
> A: We appreciate your feedback on our oversight.
>
> Weakness_5.
>
> A: The reported a_0 from the previous methods are measured after the training on the base session. However, as we have described in Sec 4.2, we evaluated performance after training on the last session.
>
> We acknowledge that the improvement in performance can be attributed to the larger backbone. We would appreciate it if the focus could shift from such a perspective to ‘exploring how to effectively utilize larger backbones (especially in the size constraint of ResNet50) in FSCIL’. (Further details are provided in the 5th response to R3)
>
> Weakness_7.
>
> A: Unlike the conventional metric, we assess session-level accuracy by exclusively evaluating the classes in a specific session only after training on the last session, i.e., the evaluation occurs solely for the classes in the i-th session to obtain a_i. Therefore, the difference(IMB) captures the extent of the disproportionately high performance of the base session compared to the average incremental performance, effectively measuring the degree of bias. Additionally, given the different composition of test set classes, our proposed 'session average' represents the average session-level accuracy, differing fundamentally from the previous one.
>
> Weakness_8.
>
> A: Here, the robustness is related to the selection of the pretrained backbone in our pipeline. As seen in Figure 4, in the case of CNNs, we notice that the supervised pretrained backbone outperforms the self-supervised ones on CUB200, close to ImageNet's distribution (compared to the other two). However, the trend reverses for Flowers102 and DCC, indicating sensitivity to the target dataset. In contrast, for ViTs, DINO consistently outperforms others across all datasets. Without the proposed datasets, choosing CNN backbones might favor the supervised pretrained one. When selecting ViT backbones, doubts may arise about DINO's performance in distributions different from ImageNet. Therefore, with our proposed datasets, we can identify backbones robust against target distribution shifts.
>
> [ref.1] Zhirong Wu et al. "Unsupervised feature learning via non-parametric instance discrimination." CVPR 2018.
>
> [ref.2] Linus Ericsson et al. "How well do self-supervised models transfer?" CVPR 2021.

---

> > ### Comment · Reviewer_SuEP · 2023-12-04
> >
> > Thanks for the response. Some of my questions are clear now. But I agree with other reviewers' comments and lean to give the decision of "below the acceptance threshold".

---

### Official Review · Reviewer_jpg9 · 2023-11-01

**Soundness:** 3 good
**Presentation:** 2 fair
**Contribution:** 2 fair
**Rating:** 5
**Confidence:** 3

**Summary:**

This paper proposed a pipeline for the incremental sessions, and used  feature extraction, classification module and prediction aggregation for the whole learning. Additionally, the paper introduced new evaluation metrics and benchmark datasets.

**Strengths:**

The method proposed in this paper is simple and the experiments are sufficient.

**Weaknesses:**

1. In the incremental sessions, whether there have two results for one sample from the NCM classifier and Lightweight Network, and how to deal with this situation.
2. In the experiments, I'm concerned about the accuracy of each session and the accuracy of each session compared to other methods.
3. Whether the paper makes the comparison without using a pre-trained model.

**Questions:**

Whether the relevant datasets can public, and please see the weaknesses

---

> ### Author Response · Authors · 2023-11-14
>
> We would like to express our sincere gratitude for the time and effort you dedicated to reviewing our paper. Our response to the weakness and questions is present below.
>
> 1. In the incremental sessions, whether there have two results for one sample from the NCM classifier and Lightweight Network, and how to deal with this situation.
>
> A: We summarize the detailed process of prediction aggregation below.
> The aggregation process primarily entails summing the two output distributions from LN and NCM. However, we noticed LN's output distribution exhibits excessive sharpness compared to NCM, leading to LN’s dominance when simply summed, making the prediction identical to using only LN's output distribution.
> To address the issue, we smooth the LN’s output (softmax) distribution with an appropriate temperature [ref.1] before adding it to that of NCM. Through multiple experiments, we determine the optimal temperature value for this adjustment. With this modification, we observed that when evaluating test data for the base session, LN's smoothened output distribution has a slightly stronger influence than that of NCM. Conversely, when evaluating test data for the incremental sessions, the influence of NCM's output distribution is slightly stronger than that of LN. Consequently, this aggregation process allows for achieving a more balanced overall performance.
> [ref.1] Zhirong Wu, Yuanjun Xiong, Stella X Yu, and Dahua Lin. Unsupervised feature learning via non-parametric instance discrimination. In Proceedings of the IEEE conference on computer vision and pattern recognition, pp. 3733–3742, 2018.
>
> 2. In the experiments, I'm concerned about the accuracy of each session and the accuracy of each session compared to other methods.
>
> A: We summarize the details about our proposed ‘accuracy of each session’ (Section 4.2) below.
> As we have described in Sec 4.2, unlike previous methods, we have conducted evaluations only after completing training on the last session. While previous approaches assess the performance for all classes encountered up to the current session after training at each session, we assess session-level adaptability by exclusively evaluating the classes relevant to a specific session only after training on the last session.
>
> If we have misunderstood your question, please feel free to clarify, and we appreciate your understanding.
>
> 3. Whether the paper makes the comparison without using a pre-trained model.
>
> A: We conducted experiments in our pipeline without utilizing a pretrained backbone (the baseline is still ALICE). The results of the experiments are as follows.
>
> | CUB200 |a_0 | Inc_avg | IMB | A_s |
> |----------|----------|----------|----------|----------|
> | w/o pretrained | 73.60 | 50.68 | 22.92 | 52.76 |
> | Ours | 79.54 | 63.77 | 15.77 | 65.20 |
>
> | Flowers102 |a_0 | Inc_avg | IMB | A_s |
> |----------|----------|----------|----------|----------|
> | w/o pretrained | 94.92 | 84.84 | 10.08 | 85.96 |
> | Ours | 96.42 | 94.11 | 2.31 | 94.37 |
>
> | DCC |a_0 | Inc_avg | IMB | A_s |
> |----------|----------|----------|----------|----------|
> | w/o pretrained | 76.94 | 44.10 | 32.84 | 49.57 |
> | Ours | 80.97 | 55.00 | 25.97 | 59.33 |
>
> The results show that using only the baseline backbone exhibits biased performance towards the base session. This reaffirms the significant assistance that the pretrained backbone provides in adapting to incremental sessions.
>
> [ref.1] Zhirong Wu et al. "Unsupervised feature learning via non-parametric instance discrimination." CVPR 2018.

---

### Meta-Review · Area_Chair_DjfR · 2023-12-08

**Metareview:**

This paper tries to mitigate the significant performance bias towards base session classes, which is widely existing in conventional methods. The study is technically solid and is verified by extensive experiments. The proposed metrics are novel and interesting. However, a main concern is the motivation and novelty of the method. Multiple reviewers mentioned that the overall pipeline is too heuristic and simple, and why the proposed pipeline is able to mitigate the bias remains not clearly demonstrated. The AC looked through the paper and agreed with the reviewers. The AC recommended a reject and believed that an improved methodology and a clearer motivation would benefit the paper a lot.

**Justification For Why Not Higher Score:**

The novelty of the proposed method is limited and why it is able to solve the bias problem is not convincing enough.

**Justification For Why Not Lower Score:**

N/A

---

### Decision · Program_Chairs · 2024-01-16

Reject